# Construction of Direct Z−Scheme SnS_2_ Quantum Dots/Conjugated Polyimide with Superior Photocarrier Separation for Enhanced Photocatalytic Performances

**DOI:** 10.3390/polym14245483

**Published:** 2022-12-14

**Authors:** Changqing Yang, Chenghai Ma, Duoping Zhang, Zhiang Luo, Meitong Zhu, Binhao Li, Yuanyuan Zhang, Jiawei Wang

**Affiliations:** School of Chemical Engineering, Qinghai University, Xining 810016, China

**Keywords:** sulfur-doped polyimide, SnS_2_ quantum dots, photocatalyst, solar light, hydrogen production

## Abstract

In this study, a novel direct Z-scheme SnS_2_ quantum dots/sulfur-doped polyimide (SQDs/SPI) photocatalyst was firstly fabricated by an in situ crystallization growth of SnS_2_ quantum dots on sulfur-doped polyimide through a facile hydrothermal method. The photocatalytic hydrogen production activity of 5SQDs/SPI samples reached 3526 μmoL g^−1^ in the coexistence of triethanolamine and methanol used as hole sacrificial agents, which is about 13 times higher than that of SPI under the same conditions and 42 times higher than that of SPI only as a hole sacrificial agent. The improvement can be related to the direct Z-scheme charge transfer in the tight interface between SQDs and SPI, which promoted rapid separation and significantly prolonged the lifetime of photoexcited carriers. The Z-scheme charge transfer mechanism was proposed. This discovery comes up with a new strategy for the development of an efficient, environmentally friendly, and sustainable sulfide quantum dots/polymer non-noble metal photocatalyst.

## 1. Introduction

Semiconductor material photocatalysts have become a hot topic of research today because of their function to convert solar energy into hydrogen energy at low economic cost [1,2] and to degrade organic pollutants in water in an environment-friendly way [3,4,5]. However, due to the low photogenerated carriers’ separation efficiency of the single-component photocatalyst, it is usually difficult to achieve highly efficient photocatalytic hydrogen production and degrade organic pollutants. Therefore, constructing a heterojunction between two semiconductors is an effective way to enhance the separation efficiency of photogenerated electrons and holes as well as improve the stability of those photocatalytic materials [6,7,8,9]. Regrettably, as a most common composite among all heterojunction material systems, type-II heterojunction can spatially separate the photogenerated charges [7]. The redox ability of the photo-induced electrons and holes are decreased in the type-II heterojunction system. This reduction in redox capacity weakens the driving force for the photocatalytic reaction of the composites. To resolve this contradiction, many scholars have devoted themselves to the research and development of a series of photocatalysts with a Z-scheme heterojunction structure [10,11,12], which can promote the separation of photogenerated carriers as well as maintain the stronger redox ability of the two semiconductor materials.

Recently, a new conjugated polyimide (PI) was successfully synthesized using a novel green method [13,14]. As a novel polymer photocatalyst, PI has many advantages, such as suitable band structure (E_g_ ≈ 2.7 eV), abundant sources, harmlessness, low cost, adjustable structure, and chemical stability. However, due to the inherent feature of the photonic excitation and carrier transport of conjugated polymer [15], PI still has a low photocatalytic activity for water reduction to produce hydrogen and degrade organic pollutants. Therefore, many research efforts are devoted to improving the photocatalytic activity of PI. These strategies include elemental doping [16,17], bandgap modulation [18], cocatalysts [19,20,21], and heterojunctions [22,23,24,25,26]. In particular, the Z-scheme CdS_2_/PI heterojunction dramatically improves the separation efficiency of photogenerated carriers and maintains a stronger hydrogen production capacity, thereby significantly enhancing its photocatalytic activity and stability [27]. Unfortunately, the high toxicity of cadmium (Cd) greatly limits its practical application. Alternatively, tin disulfide (SnS_2_) has received increasing attention because of its abundance, environmental friendliness, low cost, and low bandgap (2.0–2.4 eV) as a visible-light photocatalyst [28,29,30]. Moreover, some studies of Z-scheme photocatalysts including a direct Z-scheme g-C_3_N_4_/SnS_2_ [31], the novel 2D/3D Z-Scheme g-C_3_N_4_/SnS_2_ [32], and dual modified MoS_2_/SnS_2_ [33] have proved that SnS_2_ is a promising material to combine with inorganic or organic photocatalytic materials for improved light harvesting and better purification of the environment. It is well known that semiconductor quantum dots (QDs) materials [34] can exhibit high extinction coefficient of light absorption and size-dependent bandgap, which will improve light-harvesting performance compared to bulk semiconductors. More importantly, QDs have a large surface-to-volume ratio, which can greatly improve the surface amplitudes of photoexcited electrons and holes for photochemical reactions [35,36].

In order to improve the photocatalytic properties of polyimide (PI), hyperbranched polyimide N-oxide (PINO) photocatalyst was synthesized by Yang et al., and sulfur-doped polyimide (SPI) photocatalyst [17] was prepared by Wang et al. Although the visible light absorption and oxidation abilities of PINO and SPI were enhanced relative to PI, their photocatalytic activity still needs to be improved due to their inherent high recombination rate of photogenerated electrons and holes. PINO was synthesized by a strategy of N-site oxidation of PI, while SPI was formed by subliming sulfur (S_4_) as a sulfur source and incorporating sulfur atoms into the backbone of PI to replace some nitrogen atoms, which has better chemical stability. Moreover, in order to improve the separation efficiency of photogenerated electrons and holes, Z-scheme composite photocatalytic materials were constructed. If PINO was selected as the base material, the oxygen in PINO was likely to react with tin sulfide, making the PINO base material change and resulting in more complex products. Therefore, we chose SPI as the base material to construct the SQDs/SPI direct Z- scheme photocatalyst.

Moreover, to the best of our knowledge, there is still no study employing SnS_2_ combined with SPI to enhance its photocatalytic activity. Therefore, it is highly desirable to design and construct SnS_2_ quantum dots/sulfur-doped polyimide (SQDs/SPI) composite photocatalysts for efficient and stable photocatalytic hydrogen production and degradation of methyl orange (MO). In this work, for the first time, we constructed a novel direct Z-scheme SQDs/SPI composite photocatalytic material through in-situ crystallization growth of SQDs on the surface of SPI by a facial hydrothermal react method. Remarkably, the decoration of SQDs on the surface of SPI shows a significant enhancement in photocatalytic activity for hydrogen evolution and degradation of MO under solar light compared with bare SQDs or SPI. The enhanced photocatalytic performance and stability can be attributed to the construction of direct Z-modes between the SQDs and the SPI dense interface, which greatly facilitates the efficient separation of photogenerated electrons and holes and suppresses photo corrosion. The mechanism of efficient interface separation of carriers was confirmed by using transient PL decay spectra and ultrafast transient absorption spectra analysis. This study will provide a new perspective and idea for the design and development of a novel green, environmentally friendly, cheap, efficient, and durable sulfide quantum dot/conjugated polymer composite photocatalytic material.

## 2. Experimental Section

### 2.1. Materials and Preparation of the Photocatalyst

All materials were analytical grade reagents and used as received without further purification. Melamine (MA) and pyromellitic dianhydride (PMDA) was purchased from Shanghai Macklin Biochemical Co., Ltd. Crystalline tin tetracloride (SnCl_4_• 5H_2_O) and thioacetamide (TAA) were acquired from Shanghai Aladdin Biochemical Technology Co., Ltd. Sulfur sublimed (S_4_) was purchased from Tianjin Damao Chemical Reagent Factory.

Sulfur-doped polyimide (SPI) was prepared by the method previously reported [17]. Typically, 1.00 g of MA, 1.70 g of PMDA, and 0.90 g of S_4_ were mixed evenly and ground fully. The mixture was placed into a boat-shaped porcelain crucible and half covered with the lid to form a semi-enclosed system. Then it was put into the pipe furnace and heated up to 325 °C at a heating rate of 7 °C/min, keeping this temperature for 4 h before it was cooled to room temperature. Thus, a pale yellow powder solid was obtained, and was named as SPI after removal of the unreacted materials.

Tin sulfide quantum dots/sulfur-doped polyimide (Shorted for SQDs/SPI) composites were synthesized using a facile immersion-hydrothermal method illustrated in Figure 1. For the example of synthetic 1 wt%SnS_2_/SPI (abbreviated as 1SQDs/SPI), 0.99 g of SPI, 0.0192 g of SnCl_4_•5H_2_O, and 0.0166 g of TAA were placed in three beakers with appropriate volume respectively. Then 40mL of deionized water was added to the beaker containing SPI, and 10 mL of deionized water was added to the other two beakers, respectively. The aqueous solution of SnCl_4_•5H_2_O and the aqueous solution of TAA were added dropwise into the dispersion solution of SPI individually under the stirring conditions. After 15 min of sonication, the resulting mixture was transferred to a Teflon-lined stainless steel autoclave and heated at 140 °C for 4 h. Finally, the resultant solid was washed several times with distilled water and named as 1SQDs/SPI. In this way, SQDs/SPI composites with different amounts of tin sulfide were obtained, and named as SQDs/SPI, where x represents the weight percentage of SQDs in the sample. Similarly, the SnS_2_ sample was prepared through the reaction of SnCl_4_•5H_2_O and TAA without adding SPI into the reactant mixture. The SQDs were obtained by ultrasonically dispersing the prepared SnS_2_ sample. All the amounts of the reactants used to prepare samples and the synthetic yields are listed in Appendix A.

### 2.2. Characterization

The crystalline phase of all samples was obtained by X-ray powder diffraction (XRD; Bruker D8, Cu Ka irradiation, k = 0.15406 nm, Japan). The morphologies of photocatalysts were recorded on a vltra55 field emission scanning electron microscope (FESEM, a Carl Zeiss Gemini). The transmission electron microscopy (TEM) and high-resolution transmission electron microscopy (HRTEM) images were recorded on a JEOL JEM-2010 field emission transmission electron microscopy. Fourier transformed infrared (FTIR) spectra were obtained on a Nicolet NEXUS870 spectrometer using the KBr pellet support. The X-ray photoelectron spectroscopy (XPS) spectra of the samples were analyzed on a VG ESCALAB MKII photoelectron spectrometer with Al Ka X-ray source. The C1 s peak at 284.8 eV arising from adventitious carbon was taken as a reference for correction of all the binding energies. The UV–vis diffuse reflectance spectra (UV–vis DRS) of the prepared samples were measured in the range from 200 to 800 nm using a varian cary 500 UV–vis spectrophotometer, and were converted to absorbance spectra by the Kubelka–Munk method. The photoluminescence spectra of the photocatalysts were measured on an Edinburgh FS5 spectrofluorometer. The transient fluorescence spectra of the photo-catalysts were recorded by an FLS980 multi-function steady-state and transient fluorescence spectrometer (Edinburgh Instruments) at room temperature. Ultrafast transient absorption spectra of the samples were conducted by the femtosecond transient absorption spectrometer (TAS) with instrument model: helios (ultrafast systems manufacturer). Electron spin resonance (ESR) spectra were performed on a Bruker model EMXplus spectrometer.

### 2.3. Electrochemical Measurements

Photo-electrochemical tests were performed by using an electrochemical analyzer (Chenhua CHI 660D) and a xenon lamp-assisted three-electrode electrochemical test system. The system consists of a Pt counter electrode, an Ag/AgCl reference electrode, a working electrode, and the electrolyte solution (Na_2_SO_4_, 0.5 mol L^−1^, pH = 6.8). The working electrode was prepared on fluorine-doped tin oxide (FTO) transparent conductive film glass. Typically, 10 mg of as-prepared sample powder and 50 μL of Nafion (25%) solution were dispersed in 1 mL of ethanol solution under stirring. After ultrasonic treating for 30 min, a uniform suspension was obtained. Then, 50 μL of suspension was added dropwise onto the precleaned FTO glass surface with an area of 2.5 cm^2^, and dried at room temperature. In this way, the drop-casted film of approximately 0.5 mm thickness is uniformly covered on the FTO glass. To bond the sample to the FTO glass more firmly, the obtained working electrode was placed in an oven at 100 °C for 1 h. Electrochemical impedance spectroscopy (EIS) plots were measured at the frequency ranged from 200 kHz to 10 mHz. Mott–Schottky curves were collected under dark at 1.0 kHz frequencies, and the open circuit potential was 10 mV. The transient photocurrents measurement was performed at the applied bias potential of 0.5 V under the full-arc light irradiation.

### 2.4. Photocatalytic Performance Tests

The photocatalytic reaction was carried out in an overhead irradiation reaction vessel connected to a glass-enclosed gas circulation system. The full arc light source was a xenon lamp with a specification of XL-300 W, which was 8.2 cm away from the reaction solution. The photocatalytic hydrogen production activities of the catalysts were evaluated in a top-irradiation reaction vessel connected to a glass closed gas-circulation system. The full arc light source was a XL-300 W xenon lamp. Hydrogen production experiment was carried out by dispersing 50 mg of the catalyst powder sample in a Pyrex glass container filled with 100 mL of the mixed solution of water/hole sacrificial agent (9:1 by volume), using CH_2_CH_2_OH)_3_N and CH_3_OH as a hole sacrificial agent. At the same time, the 3 wt%Pt as a cocatalyst was photodeposited on the surface of photocatalysts using H_2_PtCl_6_ as a precursor. To facilitate the deposition of Pt, the reactant solution was first irradiated under full acr light (λ > 300 nm) for 1 h. The amount of hydrogen produced was analyzed by an online gas chromatograph (GC–14C, Shimadzu, TCD, Ar carrier). To further appraise the photostability of the samples, cycle experiments were performed for four runs, and the irradiation time of each run was 8 h. Similarly, the used sample was fully separated and collected from the solution by high-speed centrifugation and added back into the Pyrex top-irradiation vessel for the next cycle.

The Photocatalytic degradation of the MO activities of the samples was also tested in a Pyrex top-irradiation vessel. Usually, 0.2 g of the catalyst was put into the three double-layer reactor containing 100 mL of methyl orange (MO) solution (400 mg/L). It was stirred magnetically in the dark for 1 h before illumination to establish the adsorption−desorption equilibrium for MO. A certain amount of the reaction mixture (3–5 mL) was withdrawn and centrifuged to remove the photocatalyst particles at the given irradiation time intervals. Then the absorbance of the obtained clear solution was measured by a UV–vis spectrometer (Mapada UV-1800) at 464 nm. 

## 3. Results and Discussion

### 3.1. Structure and Morphology Analysis

The crystal structure of as-prepared SPI, SQDs, and SQDs/SPI composites was analyzed by powder X-ray diffraction (XRD) pattern. As shown in Figure 1, the SPI sample displayed several distinct peaks in the range of 10−30°, indicating that the three monomers of S_4_, MA, and PMDA polymerized into the framework of SPI [17]. Meanwhile, these characteristic peaks were clearly observed in a series of SQDs/SPI samples, demonstrating that the introduction of SnS_2_ did not change the original crystal structure or polymeric chain of SPI by introducing Sn species. Notably, the intensity of the diffraction peak at 29.6° of the SQDs/SPI samples diminished obviously with increasing SQDs contents. By contrast, the intensities of the peak (located at 27.3°) ascribing to the stacking of the conjugated aromatic system increased with the increase of SQDs [37]. This may be attributed to the partial stripping of the SPI surface layer under intense stirring and heating (at 140°) during the preparation of SQDs/SPI composite, resulting in an overall reduced polymerization and more significantly ordered layer-lamination stacking of SPI. The diffraction peak intensity of SQDs is amplified by six times as shown in Figure 1. 

The diffraction peaks of SQDs located at 2θ = 15.6°, 28.5°, 32.7°, and 50.1° can be indexed to (001), (100), (101), and (110) planes (JCPDS No. 23–677) [38]. These Crystal planes are also observed from the TEM images of SQDs as shown in Appendix A. However, most characteristic diffraction peaks of the SQDs cannot be observed on the XRD patterns of SQDs/SPI composites, suggesting that Sn species are highly dispersed on the surface of SPI. We speculate that the absence of the characteristic diffraction peaks of the SnS_2_ in XRD patterns of SQDs/SPI is most likely due to the very weak characteristic diffraction peak intensities, the very small amounts and ultra-small size of SnS_2_ (later verified by SEM and TEM), as well as stronger host−guest interaction that will be confirmed later by FTIR and XPS. Most of the characteristic diffraction peaks of tin sulfide could not be observed in the composites, but a careful comparison from Appendix A shows that SPI has no characteristic peak at 28.5°, while the characteristic peak of SnS_2_ at 28.5° corresponds to the (100) crystal plane. After the SQDs were in-situ crystallized on the SPI surface, the 10SQDs/SPI composite showed a weak peak at 28.5°, indicating that the tin sulfide quantum dots were indeed loaded on the SPI dominated by the (100) crystal plane, which was in agreement with the TEM results (in Figure 2).

The morphology and microstructure of the as prepared samples were characterized by TEM, HR-TEM, SAED, and EDX. As shown in Appendix A, The TEM image of SPI displays lamellar morphology. From the TEM image (Appendix A) also can be observed the nanoparticle morphology of SQDs. Appendix A illustrates the high-resolution transmission electron microscopy (HRTEM) image of the SnS_2_ sample, which shows the HRTEM lattice fringes of the SnS_2_. It can be seen that the three kinds of very clear lattice plane spacing are 0.596 nm, 0.275 nm, and 0.311 nm, corresponding to the (001), (101), and (100) planes [39], respectively. Meanwhile, the SAED of SnS_2_ reveals the polycrystalline nature and dominant (001), (101), and (110) planes as shown in Appendix A [40]. This result is in good agreement with the XRD results of the SnS_2_ as shown in Figure 1. The morphology of the composites formed by in-situ crystallization of SnS_2_ quantum dots grown on SPI was further studied. As shown in Figure 2a, many tin sulfide quantum dots are embedded on the SPI. It is worth noting that the HRTEM images of the 10SQDs/SPI sample can be observed from Figure 2b, indicating that many SQDs with clear lattice fringes are etched on the SPI. Simultaneously, these SQDs mainly expose the (100) crystal plane, corresponding with the lattice fringe spacing of 0.326 nm, which is larger than that of SnS_2_ before being combined with SPI. This may be attributed to the high dispersion of a small number of SQDs precursors on the SPI surface and the layered surface-induced effect caused by the strong interaction between SPI and SQDs host–guest. The interaxial angles of 120° can also be clearly observed from Figure 2b, and consistently assigned to hexagonal SnS_2_ [41]. Moreover, although the (100) plane-dominated SQDs are few and highly dispersed, a weak peak corresponding to the (100) plane can be seen in the XRD patterns of the 10SQDs/SPI composite (Appendix A). Interestingly, cadmium sulfide nanocrystals grow on the surface of polyimide (PI), which is also dominated by (100) crystal plane [27]. The EDX spectrum of the 10SQDs/SPI sample illustrates the obvious signals of C, N, O, Sn, and S elements as shown in Figure 2c, and the corresponding EDX mapping results further confirm the homogeneous distribution of SnS_2_ in the SPI.

The FTIR spectra of SPI and SQDs/SPI composites samples are shown in Figure 3. From the overall observation, most of the infrared characteristic absorption bands of SPI appear in the infrared spectrum of the composite material [17] and the absorption intensity of the composite material is weaker than that of SPI, which is consistent with the analysis results of XRD. The peaks at 1725 and 725 cm^−1^ (marked by blue dotted line) are attributed to the symmetric stretching and bending vibrations of the imide carbonyl groups (−C=O) in the PMDA moiety of SPI, respectively, but it is worth noting that some absorption bands have changed after introducing Sn species into the SPI. The symmetric stretching intensity of (−C=O) is obviously weaker than that of SPI. The absorption bands located at around 1639 cm^−1^ (marked by pink dotted line) assigned to H_2_O bending vibrations disappeared [42], which is due to the loss of absorbed water molecules at 140 °C. The absorption bands located at 1560cm^−1^ and 1403 cm^−1^ can be ascribed to C=N stretching mode and C−N bond [38]. These two absorption peaks even disappeared in the composite after the introduction of tin sulfide. These results may be due to the strong interaction between SQDs and SPI. Through detailed comparison, a new absorption peak around 1775 cm^−1^ is clearly observed in the SQDs/SPI composite, which may be assigned to the characteristic peak of Sn−N bond [27]. The formation of Sn−N bond was probably caused by coordination of the N atoms of heptazine units to the unsaturated Sn sites of SnS_2_ in the interface of SQDs/SPI, confirming the strong chemical interaction between SQDs and PI. This strong interaction facilitates the construction of a tight contact interface and the transport of photogenerated carriers between SQDs and SPI, leading to enhanced photocatalytic performance.

XPS technology was employed to further analyze the surface chemical state and bonding configuration of the as-prepared SPI, SnS_2_, and 10SQDs/SPI composite. The binding energies of all samples were corrected with reference to the C1s peak (at 284.6 eV) arising from the sp^2^ C=C bonds or adventitious carbon [43]. Shown in Figure 4 are the corresponding high resolution XPS spectra of C 1s, N 1s, Sn 3d, and S 2p of the photocatalysts, respectively. It can be clearly seen that the C 1s XPS spectra for SPI could be fitted into three characteristic peaks centered at 284.6, 287.5, and 288.7 as shown in Figure 4a. Besides the C 1s peak at 284.6 eV, the two peaks at 287.5 and 288.7 eV were attributed to N−C−N and C=O bonds in the triazine rings of SPI, which were slightly shifted to 287.8 and 288.9 eV in the XPS spectrum of 10SQDs/SPI, respectively [14]. 

From Figure 4b, three N1s characteristic peaks centered at 397.7, 398.7, and 399.1 eV for SPI can also be observed, respectively. The peak at 397.7 eV arises from the pyridine-like nitrogen atoms (N−C=N) in the triazine ring, and the two peaks at 398.7 and 399.1 eV could be assigned to the nitrogen atoms connecting with two carbonyl (−C=O) groups in the five-membered imide ring of SPI [17]. The associated binding energies for the SQDs/SPI hybrid all showed slight increases to 397.9, 399.0, and 399.5 eV, respectively. The increases in C 1s and N 1s binding energies of 10 SQDs/SPI composite are further evident from Appendix A, respectively.

In addition, the Sn 3d spectrum of SnS_2_ sample displays two main peaks at 487.0 and 494.8 eV, which can be attributed to Sn 3d_5/2_ and 3d_3/2_, respectively. The corresponding peak positions of SQDs/SPI show minor decreases to 486.6 and 494.5 eV as shown in Figure 4c. The S 2p high-resolution spectrum for SnS_2_ (Figure 4d) exhibits four characteristic peaks located at 160.5, 161.3, 161.8, and 163.0 eV. The binding energies at 160.5 and 161.8 eV can be associated with the S 2p_3/2_ and S 2p_1/2_ of S^2−^ species, respectively, and the peaks at 161.2 and 163.0 eV can be indexed to bridging S_2_^2−^ and apical S^2−^ species [44,45]. For the SQDs/SPI, the binding energies corresponding to the four main peaks shifted toward lower values of 160.3, 161.0, 161.5, and 162.9 eV than that of SnS_2_. Obviously, the binding energies of C 1s and N 1s showed an overall shift towards higher values after the introduction of tin sulfide quantum dots in SPI, while the SnS_2_ unit demonstrated the opposite trend. It is well known that the variation of binding energy is related to the change of surface electron density, which is caused by electron transfer between semiconductors with different E_F_ levels [46,47]. Typically, electrons will transfer from the semiconductor with a higher E_F_ level to the semiconductor with a lower E_F_ level in the process of hybridization of two semiconductors [31]. Consequently, electrons would transfer from SPI to SQDs during the hybridization, leading to a decrease of electron concentration in SPI and an increase in SnS_2_. Therefore, C 1s and N 1s signals shifted towards higher binding energy, while Sn 3d and S 2p shifted towards low binding energy for SQDs/SPI composite compared to pristine SnS_2_ and SPI. These results demonstrated the generation of an interfacial electric field (IEF) between SnS_2_ and SPI, with the orientation from SPI to SnS_2_. Moreover, the sp^3^-hybridized N atoms in tertiary nitrogen groups of g-C_3_N_4_ can provide the lone pair electrons to the unoccupied d orbital of transition metal (M) atoms to form M (δ^+^)−N (δ^−^) bonding states through chemical coupling interaction as previously reported [48]. Likewise, this interaction between SQDs and SPI matches well with the formation of Sn−N bond in FTIR spectrum of the 10SQDs/SPI composite sample. Therefore, the construction of a dense interface between polymer supports and transition metal sulfides based on this strong interaction is beneficial to the transport of carriers and the separation of photogenerated electron-hole pairs, thereby effectively enhancing their photocatalytic activity and stability.

### 3.2. Optical and Electronic Properties

The optical absorption properties of SPI, SnS_2_, and SQDs/SPI composites were investigated by UV–vis diffuse reflectance measurement as shown in Figure 5a. It can be clearly observed that the light absorption band edge of SPI is about 486 nm corresponding to a band gap of 2.55 eV, and the absorption of visible light is relatively weak. However, pure SQDs shows spectral absorbance over the entire visible light range with an *E*_g_ of 2.0 eV, which suggests that SnS_2_ is a potential visible light-responsive sensitizer (Figure 5b) [49]. Moreover, the visible light absorption of the SQDs/SPI composites increases gradually with the increase of SQDs content after in-situ crystalline growth of SnS_2_ quantum dots on SPI. Even more surprising is the obvious blue-shift of the absorption band edges of the SQDs/SPI composites relative to pure SPI, which is due to the quantum size effect of tin sulfide quantum dots [20]. In this way, the combination of the two semiconductors not only enhances visible light absorption for better utilization of solar energy, but also widens the band gap to further enhance the redox ability of photogenerated electrons and holes. The flat-band potential (*E*_fb_) of SPI and SQDs was evaluated in a three-electrode system by Mott–Schottky curves in 0.5 M Na_2_SO_4_ electrolyte. As Figure 5c displays, the positive slopes of C^−2^–E plots illustrated that both SQDs and SPI are n-type semiconductors. The flat band potential (*E*_fb_) of SPI and SQDs are −0.93 and −0.25 V versus Ag/AgCl (in Figure 4d), respectively. It is well known that the conduction band potential (*E*_CB_) for n-type semiconductors is typically about −0.1 or −0.2 eV more negative than its flat band potential [27]. Therefore, the conduction band potentials of SPI and SQDs are estimated to be −0.93 and −0.25 V relative to a common hydrogen electrode (NHE) ($E_{NHE}=E_{Ag/ACl}+0.197\;V$), respectively. Furthermore, combining the Mott–Schottky results and the band gaps, the valence band (*E*_VB_) potentials of SPI and SQDs can be calculated to be 1.62 and 1.75 eV by the equation: *E*_VB_ = *E*_CB_ + *E*_g_ [50], respectively. To further determine the *E*_VB_ potentials of SPI and SQDs, VBXPS spectral tests were performed. As displayed in Appendix A, the E_VB_ positions of SPI and SQDs are about 1.62 and 1.75 eV, respectively, indicating the E_CB_ positions of SPI (−0.93 eV) and SQDs (−0.25 eV) were reasonable. Thus the estimated *E*_CB_ and *E*_VB_ positions of SPI and SQDs can be obviously seen in Figure 5d. There is a typical type-II heterojunction structure between SPI and SQDs samples, which is very beneficial to promote the interface separation of photogenerated electrons and holes, thereby effectively improving their photocatalytic activity. 

### 3.3. Photocatalytic Performance and Mechanism

To appraise the photocatalytic performance of the prepared photocatalysts, the photocatalytic hydrogen-evolution activities of SPI, SQDs, and SQDs/SPI composite samples were tested by using CH_2_CH_2_OH)_3_N as a hole sacrificial agent under full light irradiation. 

As shown in Figure 6a, only a very small amount of hydrogen was detected under illumination when only SQDs with 3%Pt acted as photocatalysts. Moreover, it can be observed that H_2_ evolution rate from all of the samples was high at an early stage as shown in Appendix A. This similar phenomenon was also found in the research work of the composite material composed of molybdenum oxide nanosheets grown on conjugated polyimide as a photocatalyst [23]. This may be because within the first hour, the photodeposited Pt nanoparticles were completely attached to the surface of the photocatalyst and played a good role as cocatalyst. At the same time, the sacrificial reagents (CH_3_OH and TEOA) were sufficient in the initial stage of the photocatalytic reaction, which can completely consume the photogenerated holes and make as many photogenerated electrons participate in the water reduction reaction as possible. However, with the passage of reaction time, a small amount of Pt nanoparticles may fall off the surface of catalyst under severe agitation and the continuous consumption of the holes sacrificial agent resulted in a slowdown in the hydrogen production rate. Moreover, it can be clearly seen that the hydrogen production activities of the composites were significantly enhanced relative to SPI, which may be attributed to the 3%Pt load on all samples as hydrogen production cocatalyst. To prove that the Pt-cocatalyst was actually loaded onto the sample, we conducted TEM analysis on 10SQDs/SPI samples loaded with 3% Pt. As shown in Appendix A, some nanoparticles are evenly dispersed on the 1010SQDs/SPI sample. After amplifying one of the nanoparticles by a high-resolution TEM, it can be clearly observed that the interplanar spacing is 0.23 nm corresponding to the (111) crystal plane of Pt [51] and the particle sizes of Pt are about 3–4 nm. This fully indicates that the platinum cocatalyst was indeed loaded on the sample and with the increase of the content of tin sulfide quantum dots, the activities of SQDs/SPI composites were gradually enhanced. When the hydrogen production activity of the 5SQDs/SPI sample reached the highest value, the activity began to decline if the content of SQDs continued to increase. Pure SPI exhibited a lower photocatalytic hydrogen production activity of about 84 μmol g^−1^ within 8 h, while the 5SQDs/SPI composite showed the highest photocatalytic activity of 277 μmol g^−1^ among all samples, which was 3.3 times higher than that of SPI at the same condition. The greatly enhanced photocatalytic activity is attributed to the efficient separation and transport of photogenerated carriers induced by in situ crystalline growth of SQDs on SPI. As for the reduction of photocatalytic activity caused by the introduction of too many SQDs, it may be mainly caused by the weakening of light absorption by SPI subject materials due to the shading effect of light.

Photocatalytic stability is another important factor for evaluating photocatalytic performance. In order to explore the stability of the prepared samples, we performed four photocatalytic hydrogen production cycle experiments in 32 h. As can be seen from Figure 6b, the hydrogen production activity of the 10SQDs/SPI composite photocatalyst has been stable, so that it still maintains a high activity after four cycles, thus showing excellent continuous cycling stability. Such good photocatalytic hydrogen production stability originates from the strong interaction between the SPI and SQDs interface and the reduced photocorrosion caused by the efficient spatial separation of carriers.

In addition, the choice of the hole sacrificial agent has a great influence on the photocatalytic hydrogen production activity. We randomly selected 3SQDs/SPI sample in the SQDs/SPI composites for photocatalytic hydrogen production tests in the presence of different hole sacrificial agents. As displayed in Figure 6c, the photocatalytic hydrogen production activity of the 3SQDs/SPI sample was better in the presence of (CH_2_CH_2_OH)_3_N than in CH_3_OH aqueous solution. Although CH_3_OH can be used as a proton source for hydrogen production [52], CH_3_OH was used as a hole sacrificial agent in many research works on photocatalytic hydrogen production [37,53,54]. Obviously, the photocatalytic hydrogen production activity of the 3SQDs/SPI sample was still very low when only methanol was used as a hole sacrificial agent, indicating that hydrogen was not mainly derived from methanol. Several comparative experiments were also carried out. As shown in Appendix A, the 3SQDs/SPI samples showed very weak photocatalytic hydrogen production activity in both anhydrous CH_3_OH and anhydrous CH_3_OH/(CH_2_CH_2_OH)_3_N systems. Using silver nitrate (AgNO_3_) as the sacrificial electron donor reagents, in the system in which AgNO_3_ and (CH_2_CH_2_OH)_3_N coexist, hydrogen was not detected in 3SQDs/SPI sample under the same illumination. These results fully indicate that a large amount of hydrogen is mainly derived from photocatalytic water reduction. Strikingly, the photocatalytic hydrogen production activity of the 3SQDs/SPI sample was greatly enhanced in the coexistence of (CH_2_CH_2_OH)_3_N and CH_3_OH. Moreover, under the same conditions, the hydrogen production activity of 3SQDs/SPI (3526 μmol g^−1^) is about 14 times higher than that of SPI (261 μmol g^−1^) as shown in Figure 6d, and it is 42 times higher than that of SPI only (CH_2_CH_2_OH)_3_N as a hole sacrificial agent. These may be attributed to the fact that the photogenerated holes of the SQDs/SPI composite material system can be more thoroughly consumed in the coexistence of CH_3_OH and (CH_2_CH_2_OH)_3_N, resulting in more photogenerated electrons participating in the reaction of reducing water to produce hydrogen.

To further understand the photocatalytic performance of the prepared samples, we first assessed the degradation of the polyimide backbone by the generated superoxide anions. As shown in Appendix A, the degradation of polyimide was very small after 9 h under full arc light irradiation. To identify the main active species for photodegradation, ESR characterization of polyimide in aqueous solution (for DMPO-•OH) and methanol solution (for DMPO-•O_2_^−^) was also performed [55,56]. It can be clearly seen from Appendix A that almost no signals of hydroxyl radicals (•OH) and superoxide anions (•O_2_^−^) are observed under dark conditions, while weaker •OH and strong •O_2_^−^ signals appear under light conditions. These fully indicate that the main active species of photodegradation is indeed superoxide anion. To further determine whether the polyimide backbone was changed before and after photodegradation, XRD characterization was performed. Apparently, several distinct peaks in the range of 10−30° had little change after photodegradation (in Appendix A), indicating that photodegradation did not change the original crystal structure or polymeric backbone of the polyimide [13,17].

We also carried out the photocatalytic degradation of the organic pollutant methyl orange (MO) in solution (4 mg L^−1^). As shown in Figure 7a, almost no degradation of MO happened on 10SQDs/SPI in the dark even for 8 h owing to the stable structure of MO, and the photolysis of MO is negligible in the absence of photocatalyst under the light. The photocatalytic activities of different samples for MO degradation can be obtained only with the addition of photocatalysts and under simulated sunlight irradiation. The obtained activities of these samples illustrated that the degradation of MO is indeed driven by simulated sunlight. The photocatalytic activity of SQDs to degrade MO is also relatively weak. It is clear that all the SQDs/SPI hybrid composites exhibit improved photocatalytic degradation of MO activity relative to SPI. The degradation activity of the SQDs/SPI composites increases initially and then reaches a maximum at ~10.0 wt% SQD content in the composite samples. However, when the SQD content was further increased, the photocatalytic degradation activity of the SQDs/SPI composites also began to decline. This is similar to the change trend of the photocatalytic hydrogen production activity of the composites. Similarly, the photocatalytic degradation efficiencies of MO for SPI and SQDs/SPI composites samples are shown in Figure 7b. It can be clearly seen that the 10SQDs/SPI sample exhibits the highest photocatalytic degradation efficiency of MO of 98.6%, which is about 300% higher than that of pure SPI. In addition, as can be seen from Appendix A, the change trend of degradation activity of 10SQDs/SPI composite sample corresponds to its absorption edge, indicating that photocatalytic degradation activity is actually dependent on the intensity of light. 

Photoelectrochemical characterization is one of the important methods to study the photocatalytic properties of semiconductors. The transient photocurrent can directly reflect the transport and separation of photogenerated carriers. The stronger the photocurrent density indicates the more efficient the transport and separation of photogenerated electrons and holes [57]. In order to better explore the photoelectrochemical properties of the SPI and SQDs/SPI composite samples, transient photocurrent tests were carried out by switching the irradiation light on and off for several cycles. As displayed in Figure 8a, the 10SQDs/SPI composite exhibited a significantly improved photocurrent compared with individual SPI. It means that the tight binding of SPI and SQDs greatly improved the spatial separation of photo-excited electrons and hole pairs and significantly reduced the recombination of photogenerated carriers. 

In addition, the interface charge transports of the electrode materials were also characterized by the electrochemical impedance spectroscopy (EIS) Nyquist plots. The EIS Nyquist plots of SPI and 10SQDs/SPI composite in 0.5 M Na_2_SO_4_ aqueous solution can be observed from Figure 8b. It was obvious that both impedance spectra were composed of a straight line at the low-frequency region and a semicircle arc at the high frequency range for SPI and 10SQDs/SPI composite samples. Compared with SPI, the EIS Nyquist plot of the 10SQDs/SPI composite at the low-frequency region illustrated the superior electrical conductivity of the photocatalyst non-photoexcited state [58], which confirms the fast interfacial charge transfer between SQDs and SPI. The above results revealed that the SQDs/SPI composite has an exuberant effect on the separation and transfer of photoexcited carriers, which is consistent with the obvious enhancement of photocatalytic activity.

Typically, photogenerated electrons and holes generated in semiconductors after being excited by incident light are able to recombine, resulting in some energy transfer into fluorescence. Therefore, the carrier recombination efficiency and photoelectron lifetime can be examined by monitoring the fluorescence intensity and lifetime of the photocatalyst. As shown in Figure 8c, the pure SPI exhibits strong fluorescence intensity, indicating that it has a high recombination rate of photogenerated electrons and holes. However, after combining with SQDs, the fluorescence intensity of the composites decreased significantly with the increase of SQDs content. This occurrence is attributed to the efficient charge transfer between SQDs and SPI, thus leading to a great improvement in the separation efficiency of the photo-excited electron and hole pairs. Interestingly, the PL peak positions of the SQDs/SPI composites gradually blue-shifted with the increase of SQDs (Figure 8c), which can be attributed to the quantum confinement effect (QCE) of SQDs [20,59]. This blue-shift phenomenon is consistent with TEM and UV−vis characterizations. In addition, the time-resolved photoluminescence (TRPL) decay spectra have confirmed that it is very useful for the characterization of charge carrier dynamics since PL is the main pathway for the recombination and dissociation of photoexcited electrons and holes [60]. The decay curves were fitted using a two-exponential fitting method. Obviously, the decay kinetics of the 10SQDs/SPI direct heterostructure are very slow compared to the nearly overlapping fast decay curves of SQDs and SPI as displayed in Figure 8d. Furthermore, the average lifetime <$\;\tau_{av}$ > was calculated by using the equation:$\;\tau_{av}=\frac{A_{1}\tau_{1}^{2}+A_{2}\tau_{2}^{2}}{A_{1}\tau_{1}+A_{2}\tau_{2}}$ [33]. As shown in Appendix A, the average lifetime < $\tau_{av}$ > are 1.81, 3.48, and 4.51 ns for SPI, SnS_2_, and 10SQDs/SPI respectively, indicating that the excited state lifetime of the SQDs/SPI composite is longer than that of SQDs and SPI. It can also be observed that the fluorescence intensities of SQDs/SPI composites decreased with increasing the amount of SQD as shown in Appendix A. The extended lifetime may originate from the direct Z−scheme structure formed between SQDs and SPI, which allows photogenerated electrons in the conduction band (CB) of the SQDs semiconductor to jump to the higher valence band (VB) of the SPI, instead of being directly recombined with the holes in the VB of SQDs [61]. This illustrates a longer interface transport distance between SQDs and SPI, resulting in a lower recombination rate of photogenerated electrons and holes, thus improving its photocatalytic activity as shown in Figure 6 and Figure 7.

To further investigate the transport dynamics of charge carriers, ultrafast transient absorption spectra of pure SPI and 10SQDs/SPI composite were performed. Since a certain amount of photocatalyst molecules are excited to the excited state by the pump pulse in the ground state, and the ground state absorption of the probe light by the sample excited to the excited state will be less than the ground state absorption of the probe light by the sample that is not excited and still in the ground state, a series of negative ground-state bleached ∆A signals were obtained in the wavelength range from 680 to 780 nm. 

As for the positive signal, it is because the sample transitions to an excited state after absorbing the pump light, and the particles in the excited state further absorb energy and transition to a higher energy level under the action of the detection pulse, so that the detector will detect a positive excited state that absorbs ∆A signal [62]. Figure 9a,b shows the transient absorption spectra of SPI and 10SQDs/SPI samples under different delay time conditions (50 fs−2 ns). Obviously, the signals of the excited state of 10SQDs/SPI composite are stronger than that of SPI, indicating that the 10SQDs/SPI sample excited to the excited state has more ground state absorption of the probe light than that of the SPI sample. Moreover, the decay curves of SPI and 10SQDs/SPI samples were fitted by two-exponential decay as shown in Figure 9c–f. At the detection wavelength of 699 nm, the carrier decay lifetime of 10SQDs/SPI composite is 388.8 ns, which is longer than that of A (58.2 ps). Likewise, the carrier decay lifetime of 10SQDs/SPI (1257.0 ps) is also longer than that of SPI (595.0 ps) at the detection wavelength of 723 nm. The prolonged decay lifetime is consistent with transient fluorescence characterization. These results provide conclusive evidence that the Sn−N bond as well as the heterojunction between SQDs and SPI can facilitate interfacial charge transfer, thereby effectively suppressing the recombination of photogenerated electrons and holes [63]. 

It is well known that photogenerated electrons can reduce oxygen to generate superoxide radicals (•O_2_^−^), and photogenerated holes react with water to generate hydroxyl radicals (•OH). Both •O_2_^−^ and •OH are the main active species for photocatalytic degradation of organic pollutants. In order to further explore the transport and separation mechanism of photogenerated carriers in the SQDs/SPI material system, it is convenient to use the ESR characterization method to test the active species of •O_2_^−^ and •OH in the photocatalytic degradation process [64]. Therefore, choosing the 10SQDs/SPI sample with the most excellent photocatalytic activity can more clearly study the generation and transport mechanism of photogenerated carriers. Thus the ESR characterization was performed in aqueous solution (for DMPO-•OH) and methanol solution (for DMPO-•O_2_^−^) [65], respectively. As shown in Figure 10a, no ESR signals of DMPO-•OH and DMPO-•O_2_^−^ were observed for the SQDs, SPI, and 10SQDs/SPI hybrid samples in the absence of light. It can be clearly seen from Figure 10b that a strong signal of DMPO-•O_2_^−^ and a very weak signal of DMPO-•OH for the SPI sample are detected under illumination.

Under the same lighting conditions, the ESR signals of DMPO-•OH and DMPO-•O_2_^−^ for SQDs are extremely weak as displayed in Figure 10c. Surprisingly, as can be seen from Figure 10d, the SQDs/SPI composite sample showed very strong DMPO-•OH and DMPO-•O_2_^−^ signals under the same illumination conditions [66]. These obtained results can be explained in conjunction with the bandgap structures of SQDs and SPI samples. It can be seen from Figure 10e that the conduction band (CB) bottom position of SPI (at −0.93 eV) is more negative than the reduction potential of superoxide radical (at −0.33 eV), so that its photogenerated electrons have enough driving force to reduce O_2_ to generate •O_2_^−^. At the same time, the valence band (VB) position of SPI is high, so its photogenerated holes have a poor ability to oxidize water to generate hydroxyl radicals, resulting in a very weak signal of the detected DMPO-•OH (in Figure 10b). However, the CB bottom position of SQDs is −0.25 eV, which is not negative enough to achieve the driving force for photogenerated electrons to combine with oxygen to generate •O_2_^−^, so that the signal of DMPO-•O_2_^−^ cannot be detected (in Figure 10c). Therefore, if the photogenerated electrons are transferred according to the traditional type II, the electrons transferred from the CB of SPI to the CB of SQDs do not have enough driving force to reduce O_2_ to generate •O_2_^−^. In fact, a strong signal of DMPO-•O_2_^−^ was detected in the 10SQDs/SPI hybrid material (in Figure 10d). These experimental results strongly demonstrated that the photogenerated electrons in the CB of SQDs directly recombine with the photogenerated holes in the VB of SPI through the Z-scheme, making more photogenerated electrons reduce O_2_ in the SPI conduction band to generate •O_2_^−^ while a large number of photogenerated holes left in VB of SQDs oxidize water to generate many hydroxyl radicals, which can be confirmed by the strong signal of the detected DMPO-•OH (in Figure 10d).

Based on the above discussion and analysis, we also proposed a photocatalytic hydrogen production mechanism as shown in Figure 2. Under full-arc irradiation, the photogenerated electrons in CB of tin sulfide directly recombine with the photogenerated holes in VB of SPI through the dense interface between SQDs and SPI. A large number of electrons in CB of SPI reduce H_2_O to generate H_2_, while triethanolamine (TEOA) as the sacrificial reagent is oxidized by the holes left in VB of SQDs [67,68]. In this way, the construction of Z-scheme heterojunction structure based on strong interaction effectively improves the transport and separation of photogenerated electrons and holes, resulting in a greatly improved photocatalytic performance.

## 4. Conclusions

In summary, a novel direct Z-scheme SnS_2_ quantum dots/sulfur-doped polyimide photocatalyst was prepared by in situ crystallization growth of SnS_2_ quantum dots on sulfur-doped polyimide through a facile hydrothermal method. Due to the strong interfacial interaction, SnS_2_ quantum dots are etched on the SPI dominated by the (100) crystal plane. The SQDs/SPI composite exhibited better catalytic ability for simulating sunlight hydrogen production and dye degradation than those of pure SPI. Transient PL decay spectra, ultrafast transient absorption spectra, and electron spin resonance spectra analysis confirmed that the photocatalytic performances of the composite were significantly enhanced by extending the lifetime of photogenerated holes and electrons and increasing its redox ability through the Z-scheme charge–carrier transfer mechanism of the SQDs/SPI photocatalyst. This study highlights the role of the strong interfacial interaction in constructing an efficient, environmentally friendly and sustainable sulfide/polymer heterojunction photocatalyst.

## Data Availability

Data will be available on request.

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
