# Peer review of "Construction of Direct Z−Scheme SnS2 Quantum Dots/Conjugated Polyimide with Superior Photocarrier Separation for Enhanced Photocatalytic Performances"

_polymers, 2022, doi:10.3390/polym14245483_

Round 1

Reviewer 1 Report

This paper reported by C. Ma et al discusses the preparation, structure and photocatalytic activity of SPI/SnS2 composite materials that were simply prepared by hydrothermal technique. Their characterization revealed that the SnS2 nanocrystals (SQDs) were dispersed on the surface of SPI polymer network, suggesting the tight interaction between these two materials. Interestingly, the photocatalytic H2 evolution activity strongly depended on the ratio of SQDs/SPI, and the authors found that the best performance was obtained by using 5SQDs/SPI. These findings are potentially interested for the readers of Polymers, but I found there are several issues in this manuscript (see below) that should be addressed before publication. Thus, at this present, major revision(s) is necessary.

1.       Experimental section, no information about the synthetic yield was shown. These should be added to Table S1.

2.       For H2 production experiments, 3wt%Pt cocatalyst was loaded on each SQDs/SPI composite. But no characterization for the Pt-loaded samples were done. The location of Pt cocatalyst is crucial for H2 evolution, TEM analysis is strongly required.

3.       Although the authors insisted that H2 evolution rate from all of the samples was high at an early stage, no data about this point is shown. Similarly, the authors also insisted that hydrogen production activity of the composite was significantly enhanced relative to SPI after loading 3%Pt, no data was shown.

4.       P13 line 14 from the bottom, the authors insisted that a large amount of hydrogen is mainly derived from photocatalytic water splitting. But, in this reaction, sacrificial electron donors (MeOH, TEOA) were used. Thus, this reaction should not be water splitting, but be simple proton reduction.

5.       Figure 8(c), the excitation wavelength for PL spectral measurements should be described in the figure caption.

6.       The data shown in inset table of Figure 8(d) are not easy to read. These data should be separately shown from this figure.

7.       Although the authors insisted that the decrease of the fluorescence intensity of SPI by increasing the amount of SQD was attributed to the efficient charge transfer between SQD and SPI, the light shielding by SQD may be another possible origin. Thus, the other SQDs/SPI composites should be evaluated by this technique.

8.       As shown Figure 10(e), the authors insisted that the two-step light excitation was required for this photocatalytic reaction. In this case, the power of the light source is a crucial factor for photocatalytic activity. Thus, the excitation light power dependence of photocatalytic activity should be carefully evaluated.

Author Response

Dear reviewer,

Thank you very much for your review about  our manuscript (polymers-2024420) entitled “Construction of direct Z−scheme SnS2 quantum dots/conjugated polyimide with superior photocarrier separation for enhanced photocatalytic performances”. We greatly appreciate the valuable comments from you.

I am sending our  the “responses to the referee” profiles. We have made the corrections according to the your comments and marked them with blue color in the revision. We have provided a point by point reply to your comment and uploaded it as a Word file.

In the revised manuscript, the following changes have been made:

1) Reference Formatting: All the references have been formatted according to journal standards.

2) We have provided 2 copies of the final manuscript file:

  1. a) One is the revised manuscript file that does not contain any highlighting or editing marks. (as the primary manuscript document file for publication).
  2. b) Another one is the marked copy of the revised manuscript that shows changes made on revision clearly highlighted. (as the Supporting Information for Review).

I hope this revised manuscript has been satisfactorily improved and hope it will be accepted for publication. We are looking forward to receiving a favorable response from you. Thank you very much for your help.

Yours sincerely,

Chenghai Ma

Author Response

Dear Reviewer,

Thank you very much for your review about our manuscript (polymers-2024420) entitled “Construction of direct Z−scheme SnS2 quantum dots/conjugated polyimide with superior photocarrier separation for enhanced photocatalytic performances”. We greatly appreciate the valuable comments from you.

I am sending our  the “responses to the referee” profiles. We have made the corrections according to the your comments and marked them with blue color in the revision. We have provided a point by point reply to your comment and uploaded it as a Word file.

I hope this revised manuscript has been satisfactorily improved and hope it will be accepted for publication. 

Yours sincerely,

Chenghai Ma

Round 2

Reviewer 1 Report

This is my second review of the paper reported by C. Ma et al about the photocatalytic performance of SnS2 QD-polyimide composite material. In this revision, the authors kindly revised their manuscript and most of my previous concerns were adequately addressed by several newly added experimental results. Thus, the paper is now almost acceptable for publication, but the following two points should be taken into consideration.

1.       In the response to my previous comments No.4, the authors still insisted that the evolved hydrogen is mainly derived from photocatalytic water splitting in spite of using sacrificial reagents (CH3OH and TEOA). Water splitting reaction must generate two molecules of H2 and one molecule of O2 from two molecules of H2O (2H2O -> 2H2 + O2). But in this paper, no O2 evolution was confirmed and only the half reaction of water splitting, that is, H+ were photocatalytically reduced to generate H2. I recommend to the authors to replace “water splitting” by “water reduction” or “proton reduction”.

2.       As shown in newly added Figure 2, almost all the samples showed comparable photocatalytic activity for the initial 1 h reaction, but the reason was hardly discussed.

Author Response

Dear reviewer,

Thank you very much for your valuable comments about our manuscript (polymers-2024420) entitled “Construction of direct Z−scheme SnS2 quantum dots/conjugated polyimide with superior photocarrier separation for enhanced photocatalytic performances”.

I am sending our “responses to the referee” profile. We have made the corrections according to the reviewer’s comments and marked them with blue color in the revision. The responses are summarized in separate sheets and uploaded to you by online submission.

I hope this revised manuscript has been satisfactorily improved and hope it will be accepted for publication. We are looking forward to receiving a favorable response from you. Thank you again for your help.

Best wishes,

Chenghai Ma

Reviewer 2 Report

the present form of the manuscript is accepptable and no further comment

Author Response

Dear reviewer,

Thank you very much for your valuable comments about our manuscript (polymers-2024420) entitled “Construction of direct Z−scheme SnS2 quantum dots/conjugated polyimide with superior photocarrier separation for enhanced photocatalytic performances”.

Thank you again for your help.

Best wishes,

Chenghai Ma